# Patterns in acute aortic dissection and a connection to meteorological conditions in Germany

Stephan Dominik Kurz[1,2☯*], Holger Mahlke[3,4☯], Kathrin Graw[5,6☯], Paul Prasse[7‡], Volkmar Falk[1,2,8,9‡], Christoph Knosalla[1,2,8‡], Andreas Matzarakis[5,6‡]

1 Department of Cardiothoracic and Vascular Surgery, Deutsches Herzzentrum der Charité (DHZC), Berlin, Germany, 2 Charité - Universitätsmedizin Berlin, Corporate Member of Freie Universität Berlin and Humboldt-Universität zu Berlin, Berlin, Germany, 3 Wetter3.de - R. Behrendt und H. Mahlke GbR, Wehrheim im Taunus, Germany, 4 Institute of Meteorology and Climate Research, Karlsruhe Institute of Technology (KIT), Karlsruhe, Germany, 5 Research Centre Human Biometeorology, German Meteorological Service, Freiburg, Germany, 6 Chair of Environmental Meteorology, Faculty of Environment and Natural Resources, Albert-Ludwigs-University, Freiburg, Germany, 7 Department of Computer Science, University of Potsdam, Potsdam, Germany, 8 German Center for Cardiovascular Research (DZHK), Partner Site Berlin, Berlin, Germany, 9 Department of Health Sciences and Technology, Translational Cardiovascular Technologies, Institute of Translational Medicine, Swiss Federal Institute of Technology (ETH) Zurich, Zurich, Switzerland

☯ These authors contributed equally to this work.
‡ PP, VF, CK, and AM also contributed equally to this work.
* stephan.kurz@dhzc-charite.de

**Data Availability Statement:** The original datasets can be accessed as follows: 1. Dataset from Forschungsdatenzentrum der Statistischen Ämter der Länder: Interested researchers can obtain the

## Abstract

Acute type A aortic dissection (ATAAD) is a dramatic emergency exhibiting a mortality of 50% within the first 48 hours if not operated. This study found an absolute value of cosine-like seasonal variation pattern for Germany with significantly fewer ATAAD events (Wilcoxon test) for the warm months of June, July, and August from 2005 to 2015. Many studies suspect a connection between ATAAD events and weather conditions. Using ERA5 reanalysis data and an objective weather type classification in a contingency table approach showed that for Germany, significantly more ATAAD events occurred during lower temperatures (by about 4.8 K), lower water vapor pressure (by about 2.6 hPa), and prevailing wind patterns from the northeast. In addition, we used data from a classification scheme for human-biometeorological weather conditions which was not used before in ATAAD studies. For the German region of Berlin and Brandenburg, for 2006 to 2019, the proportion of days with ATAAD events during weather conditions favoring hypertension (cold air advection, in the center of a cyclone, conditions with cold stress or thermal comfort) was significantly increased by 13% (Chi-squared test for difference of proportions). In contrast, the proportion was decreased by 19% for conditions associated with a higher risk for patients with hypotension and therefore a lower risk for patients with hypertension (warm air advection ahead of warm fronts, conditions with no thermal stress or heat stress, in the center of a cyclone with thermal stress). As many studies have shown that hypertension is a risk factor for ATAAD, our findings support the hypothesized relation between ATAAD and hypertension-favoring weather conditions.

dataset from the Berlin location (Standort Berlin) of the Forschungsdatenzentrum der Statistischen Ämter der Länder. For further details on accessing data, please visit their official website at http://www.forschungsdatenzentrum.de or contact them directly via email at forschungsdatenzentrum@statistik-bbb.de. 2. Dataset from Deutsches Herzzentrum Berlin: The dataset can be obtained by reaching out to Deutsches Herzzentrum Berlin. For more information or specific inquiries about the data, visit their official website at www.dhzb.de or contact them via email at typa@dhzc-charite.de."

**Funding:** The authors received no specific funding for this work.

**Competing interests:** The authors have declared that no competing interests exist.

## Introduction

Acute type A aortic dissection (ATAAD) is a medical emergency with high mortality rates if not operated. An aortic dissection is a separation of the aortic wall layers. Most often caused by an intimal tear in the medial aortic wall layer through which blood penetrates into the aortic wall. Abrupt onset of severe chest pain is a frequent symptom of ATAAD, while type B dissections are more frequently associated with pain in the back or abdomen. The mortality of ATAAD is 50% within the first 48 hours. That is double the mortality of patients with acute Type B [1]. The incidence of the disease ranges from 4.6 to 15 cases per 100,000 inhabitants in Germany [2–4]. The German Heart Center Berlin established an "Aortentelefon" (aortic emergency hotline) to provide quicker response times and sharper diagnostic accuracy for patients with ATAAD [5]. The first contact with a medical facility for ATAAD patients is usually the ambulance service or the emergency room [6]. A study by [7] found that emergency surgery for ATAAD is associated with a high risk for post-traumatic stress disorder (PTSD) negatively affecting physical and mental well-being. Efforts should be directed at prevention, early diagnosis, and therapy of PTSD in these patients [7]. A study by [8] aimed to determine the incidence of ATAAD in emergency departments and identified that approximately 50% of ATAAD cases may remain undetected [8].

Temporal patterns of ATAAD events have been identified, with a higher number of events occurring in winter and a lower number in summer [9–24]. Several studies have also examined the association between ATAAD events and weather conditions, such as short-term temperature changes and diurnal temperature range [13–15, 20, 24–35], as well as air pollution [36]. [37] summarized the results from [35] for China as a recent overview of the association between ATAAD events and weather conditions.

Previous studies have identified hypertension as a significant risk factor for ATAAD [22]. Elevated blood pressure increases the tension in the arterial wall, which can increase the risk of tearing [1]. Seasonal variations in blood pressure have also been documented, with higher blood pressure levels observed during colder months [38]. As males have a higher prevalence of hypertension than females, male individuals are at a higher cardiovascular risk for aortic dissection [39–42]. Moreover, the incidence of thoracic aortic diseases is found to be higher in men than in women, with a rate of 16.3 cases per 100,000 per year for men and 9.1 cases per 100,000 per year for women in the Swedish population [43].

Studies have also demonstrated a relationship between temperature and blood pressure, especially in elderly individuals [44–47]. Cold temperatures have been found to trigger aortic dissection events [22, 32, 48]. [49] reported that a decrease of 5 C in maximum temperature in Germany was associated with a 3% increase in the risk of hospital admission for hypertensive disease. The mechanism influencing blood pressure depending on temperature is vasodilation and constriction. During low temperatures, the outer blood vessels constrict so that the blood pressure rises. During high temperatures, the outer blood vessels expand causing the blood pressure to decrease.

Although the association between weather conditions and aortic dissection has been studied in Germany [50–52], it has not been thoroughly examined. This topic is of interest to human-biometeorology researchers in Germany, as it involves understanding the complex interactions between the atmospheric environment and human physiology [53]. While the physiological response to weather conditions varies depending on an individual's adaptive capacity and health status, establishing causal relationships between these factors is challenging [54, 55].

The present study aims to investigate the possible correlation between meteorological conditions and the risk of aortic dissection in Germany, using anonymized patient data from the

German Federal Statistical Office and the German Heart Center, Berlin. We will examine the weather conditions and meteorological parameters significantly associated with the occurrence of aortic dissection events in Germany and explore the hypothesis that colder temperatures are associated with a higher risk of aortic dissection events. The study will utilize the ERA5 high-resolution atmospheric reanalysis data for total Germany and bio-synop class data (see section "Materials and methods") provided by the German Meteorological Service (DWD) for the region of Berlin and Brandenburg originally based on the bio-synop classes of [56].

## Materials and methods

### Data

**ATAAD data.** This observational study is based on two data sets. One data set contains data for the whole of Germany and was collected and provided by the German Federal Statistical Office (Statistisches Bundesamt), Wiesbaden. It is based on anonymized medical diagnosis data provided by physicians and hospitals practicing in Germany (as seen in [57]). Their medical diagnoses are coded using the ICD-10-GM key (International Classification of Diseases and Related Health Problems, 10th revision, German modification [ICD-10-GM]). Each aortic dissection event (Type A and Type B) is reported along with diagnosis (ICD codes I71.00-I71.07, dissections) and date. Due to restrictions provided by German data protection laws, the data set is anonymized, and the locations of reported ATAAD events are not provided in the data set. Moreover, data are reported as sums of five days periods. The data set covers the period 2005 to 2015 and comprises 53,176 cases in Germany. For more insight into the data set, see [57].

The other data set included the ATAAD events that happened for the region of Berlin and Brandenburg from 2006 to 2019 (978 cases) and was collected by the German Heart Institute Berlin (DHZB) based on hospital admissions with the diagnoses ATAAD (ICD codes I71.00 -I71.07). The data set was anonymized and included the gender and age of patients and the onset time of pain.

**ERA5 atmospheric reanalysis data.** The first meteorological data set of selected parameters was compiled from the ERA5 atmospheric reanalysis data set which is provided by the European Centre for Medium-Range Weather Forecasts (ECMWF) from the European Copernicus Program [58, 59]. It has a horizontal resolution of 0.25 degrees and temporal resolution of 1 h. As the region of study is Germany, average values were computed for the area of Germany. It should be noted that this might smooth out important phenomena such as cold fronts and local effects.

**Objective weather type classification.** The objective weather type classification of the DWD [60, 61] was used. For each day, it classifies the weather pattern predominant in Germany based on the prevailing wind in the lower troposphere (700 hPa), the cyclonality in the lower and the mid-level troposphere(1000 hPa and 550 hPa), and the humidity of the troposphere. There are 40 different weather types, and a numerical index denotes each.

**Bio-synop classes.** Another method for objective classification of weather from a biosynoptical point of view was developed by [56]. The so-called bio-synop classes characterize the occurrence of certain effects on human health induced by meteorological factors with the help of statistical correlations from epidemiological studies [53]. The DWD routinely applies 5 bio-synop classes to predict effects on human health for people who are weather-sensitive or weather prone. In Germany, the weather affects the health of approximately every second inhabitant according to several representative population surveys, the latest in 2021 [62]. The bio-synop classes consist of a combination of three values [63]. The first value (numbers between 1 and 5) describes a weather condition:

- 1 implies weather conditions in the center of a high-pressure area

- 2 implies weather conditions in front of a warm front (warm air advection)

- 3 implies weather conditions in the center of a cyclone

- 4 implies weather conditions behind a cold front (cold air advection)

- 5 implies neutral weather conditions (no condition from 1—4 is appropriate)

The second value describes the intensity of the weather conditions, in particular the intensity of the changing weather:

- 0 implies no change

- 1 implies some change

- 2 implies a very strong and often fast change in weather

The third value describes thermal stress:

- 00 for thermal comfort

- k1, k2, k3 for slight to strong cold stress

- w1, w2, w3, w4 for slight to very strong heat stress

The bio-synop classes predicted by DWD for the current day and two subsequent days are available since 2002 for 11 areas in Germany and are divided into two halves of the day. For the analysis, the bio-synop classes for the region of Berlin and Brandenburg for the second half of the current day have been used together with ATAAD events of the following 24 hours. The time period regarded was 2006 to 2019.

## Methods

a) The Wald-Wolfowitz runs test is a non-parametric test for randomness of time series of categorical variables [64–66].

b) Box plots are a graphical method of representing data showing the range, quartiles, whiskers (1.5 times the interquartile range from the median), and outliers of a sample of data.

c) The restrictions provided by the German data protection laws, resulting in the fact that daily sums for the whole of Germany only were provided and the presence of other, non-meteorological factors that manifest in noise required discretization of the data concerning the number of ATAAD events and a contingency table approach. The days studied were categorized into three groups:

- group (-1): days when a low number of ATAAD events occurred. By subgrouping all days when a number of ATAAD events less than the first quartile occurred.

- group (0): days with an average number of ATAAD events. By subgrouping the days that fell into the interval between the two middle quartiles.

- group (+1): days with a large number of ATAAD events. By subgrouping all days with more than the third quartile of ATAAD events.

Then, for the continuous meteorological data, we assessed the differences in meteorological parameters between group (-1) and group (+1) using the Wilcoxon rank sum test [67, 68], which is a non-parametrical test for the difference between two populations under the null hypothesis that there is no difference.

For the categorical meteorological data, after categorizing the days into the groups mentioned above, for each characteristic value of a categorical parameter, e.g., a weather class $c$, we computed the proportion of each group by the total number of days, $P_c$. Each such proportion was compared to the overall proportion of days with a minimum of one ATAAD event to the total number of days, using the Chi-squared test for the difference of two proportions to check whether the differences were significant.

d) For the Berlin data set, an analogous approach was used. For each bio-synop class, the proportion of days with a minimum of one ATAAD event by the total number of events was determined. In order to reveal relationships between bio-synop classes and ATAAD events, these proportions were compared to the proportion of days with a minimum of one ATAAD event for the whole period. The above-mentioned Chi-squared test was used to check for statistical significance. The calculations showed statistical relationships when connecting the synop-classes predicted for the second half of the day with the occurrence of ATAAD events during the subsequent 24 h. In a last step, this allowed for compiling two combinations of bio-synop classes that showed a more frequent occurrence of ATAAD events and a less frequent occurrence.

## Results

### Trend

Based on the Germany-wide data set, in the period 2005 to 2015, 53,176 ATAAD events were recorded. Annually, this is, on average, 4,834 ATAAD events for the period given. Accompanying, the time series of the mean temperature (monthly mean values) for this short period indicates no trend (signal-to-noise ratio = 0.13, $p$ = 0.445, data are approximately normally distributed). Regarding the monthly numbers of ATAAD events for Germany, the data set shows a significant upward trend (Fig 1; signal-to-noise ratio = 2.62, $p$ = 0.009, data are approximately normally distributed). A similar trend was observed for the Berlin data set. This may not imply an actual increase in the number of ATAAD cases. Instead, the trend is attributed to a heightened awareness of ATAADs in emergency medicine and advancements in examination methods due to the improved availability of diagnostic tools (e.g., ECG-gated CT angiography) and counseling services (such as the Aortentelefon, see Introduction section and [5]). The monthly values were weighted by each month's length, and the linear

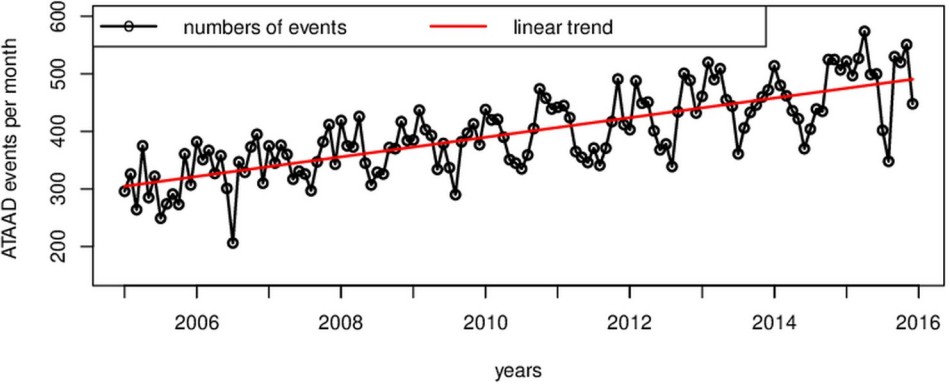

**Fig 1. ATAAD events by months.** Time series of monthly numbers of ATAAD events in Germany and linear trend line (red). The data points were weighted by the length of each month. There is a significant upward trend (signal-to-noise ratio = 2.62, $p$ = 0.009, data are approximately normally distributed).

trend was removed. The adjusted data sets were used in the calculations presented in this study.

## Risk group

For the Berlin data, a proportion of 83.0% of all recorded ATAAD patients suffered from hypertension. In Germany, the prevalence of hypertension is 55.3% [69], so hypertension and ATAAD do not occur independently, and there is a positive correlation (Chi-squared test with $\chi^2 = 343.8$, $df = 1$, $p < 0.001$) confirming earlier studies that showed that high blood pressure is a risk factor for ATAADs (see Introduction section). Patients with hypertension constitute a risk group for ATAAD.

## Seasonal patterns

There is statistical evidence that the ATAAD data for Germany show non-random patterns (runs test statistic = −2.480, $p = 0.013$). This shows that there is a tendency for days with a large number of ATAAD events to occur in series.

The recorded ATAAD events for Germany show a clear seasonal variation pattern with minima in summer and maxima in spring, fall, and winter (Fig 2). The trigonometrical pattern resembles the absolute value function of cosine (Fig 2, red curve). After the removal of the trend, significantly fewer ATAAD events occurred in summer than in the other seasons: On average, 1334 ATAAD events occurred during summer (June, July, August) per year, while an average of between 1488 and 1544 ATAAD events occurred in the other seasons. There is convincing statistical evidence that the number of ATAAD events in summer differs from those in the other seasons (two-sided Wilcoxon rank-sum test with $W = 1$, $p < 0.001$). The boxplot also illustrates this difference, showing that the summer distribution does not overlap the others (Fig 3). This clearly visible result of Fig 3 is slightly different in the monthly representation (Fig 4). Although a similar pattern is visible in the monthly representation, Fig 4 reveals a decreased number of ATTAD events in May and September, as well, and a higher amplitude is visible in April and October compared to the winter months. At this stage of our research, we cannot explain the minimum of ATAAD events in December where presumably non-meteorological reasons for a drop of reported ATAAD cases in the last third of December prevail. The region of Berlin and Brandenburg depicts a similar pattern (Fig 5) and significantly fewer cases in the summer months (two-sided Wilcoxon rank-sum test with $W = 196$, $p = 0.032$).

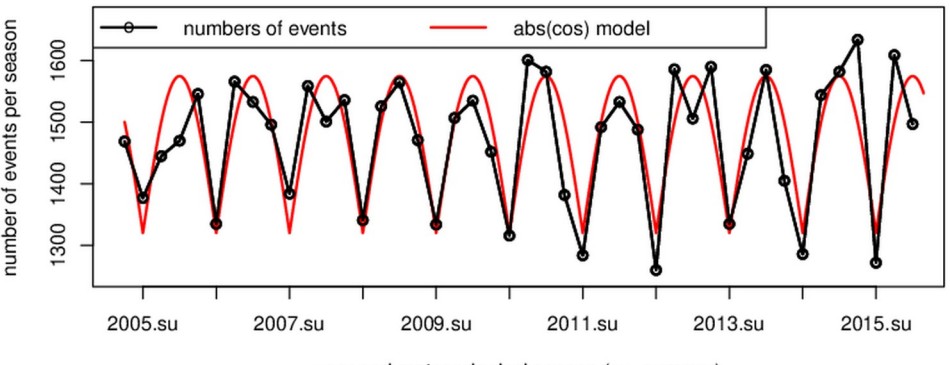

**Fig 2. ATAAD events by season.** Numbers of ATAAD events per season (su: summer) recorded for Germany (black circles). Data were fitted to a trigonometric function: absolute value of cosine (red).

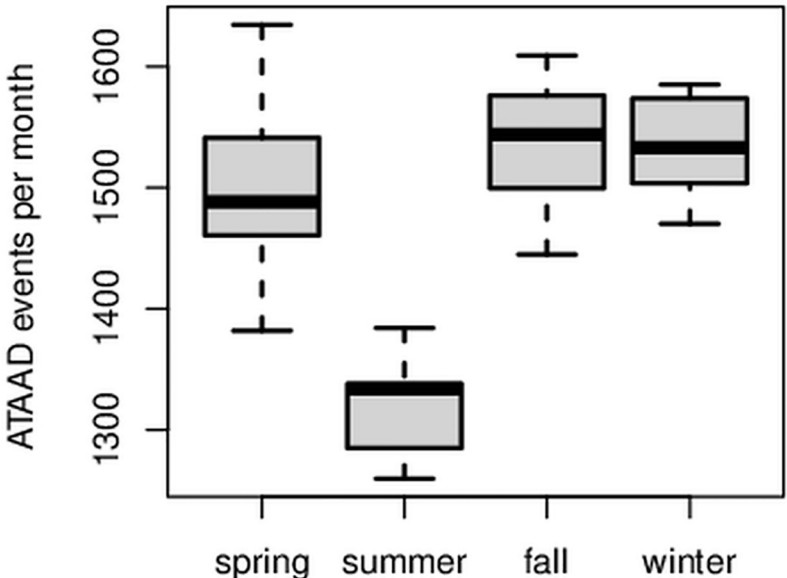

**Fig 3. Comparing ATAAD events by seasons.** Box plots for ATAAD events recorded in Germany, 2005 to 2015, per meteorological seasons (spring = March/April/May, summer = June/July/August, etc). The summer exhibits less events.

## Association of ATAAD events and meteorological parameters

We identified the differences between those days when a large number of ATAAD events occurred and a small number occurred (see section "Methods"). Because in this study, only numbers for the whole of Germany were available, area means of meteorological parameters

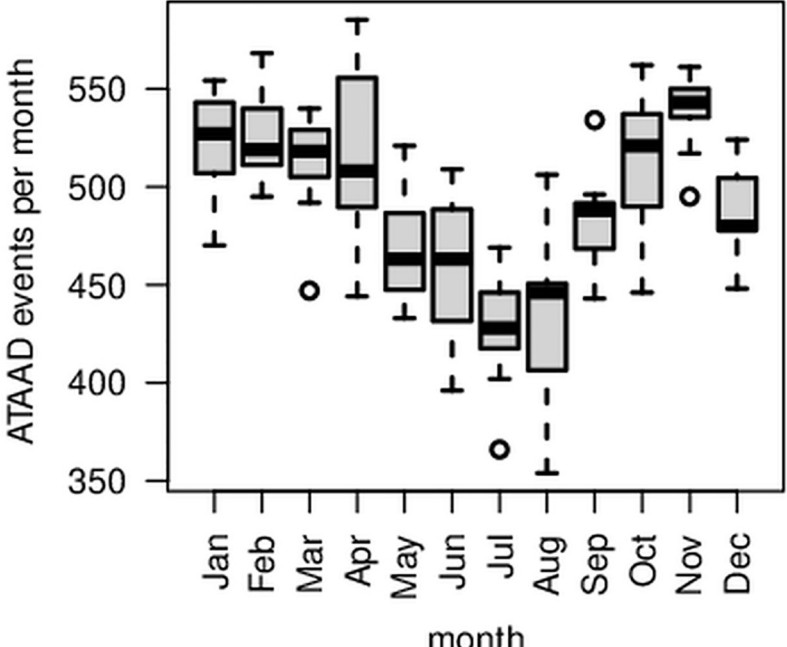

**Fig 4. Comparing ATAAD events by months.** Box plots for ATAAD events recorded in Germany, 2005 to 2015, per month. The warm months (May to August) exhibit less events.

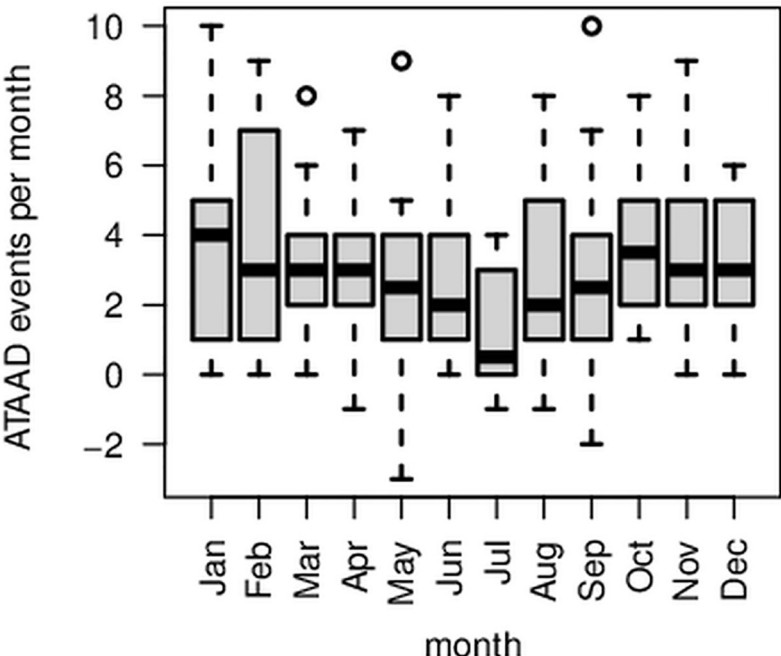

**Fig 5. Comparing ATAAD events by months.** Box plots for ATAAD events recorded in the region of Berlin and Brandenburg (2006 to 2019), per month. Negative values result from removing the trend.

were used to analyze associations with the frequency of occurrence of ATAAD events. There is significant evidence that for those days with a large number of ATAAD events, the mean 2 m temperature was lower (by about 4.8 K), and the mean water vapor pressure (2 m) was less (by about 2.6 hPa) (Table 1).

## Association of ATAAD events and weather patterns

Wind direction is associated with the advection direction of air masses and is an essential factor for discussing large-scale weather patterns. In the Northern Hemisphere, the mid-latitude temperatures are often low if NE winds prevail whereas SW winds prevail during warm periods. From the contingency table (Table 2), the occurrence of a large number of ATAAD events

**Table 1. Comparing temperature and water vapor pressure for days with different frequencies of ATAAD events.**

| | Days when many ATAAD events happened (subscript: 1) | Days with an average number of ATAAD events (subscript: 0) | Days when less ATAAD events happened (subscript: −1) | Hypothesis test | |
|---|---|---|---|---|---|
| Temperature $T$ (in 2 m) | $\bar{T}_1$ : 279.6 K | $\bar{T}_0$ : 282.1 K | $\bar{T}_{-1}$ : 286.6 K | Hypotheses: H$_0$: no difference H$_A$: $T_1 < (T_0$ & $T_{-1})$ | Result: significant. ($W = 980634.5$, $p < 0.001$) |
| Water vapor pressure $e$ (in 2 m) | $\bar{e}_1$ : 8.2 hPa | $\bar{e}_0$ : 9.6 hPa | $\bar{e}_{-1}$ : 12.0 hPa | Hypotheses: H$_0$: no difference H$_A$: $e_1 < (e_0$ & $e_{-1})$ | Result: significant. ($W = 997284.5$, $p < 0.001$) |

This table compares the mean values of temperature and water vapor pressure (based on ERA5 reanalysis data) for days where a small number of ATAAD events, an average number of ATAAD events, and a large number of ATAAD events (definition: see text) happened. Also: The results of a hypothesis test (null hypothesis H$_0$: no difference; one-sided alternative hypothesis H$_A$: difference, level of significance $\alpha = 0.05$).

**Table 2. Contingency table comparing frequency of ATAAD events and large-scale wind directions.**

| | Number of days | | | total | Proportion $P_c$ | | | Ratio factor $P_c/P$ | | |
|---|---|---|---|---|---|---|---|---|---|---|
| | low (-1) | medium (0) | large (+1) | total | low (-1) | medium (0) | large (+1) | low (-1) | medium (0) | large (+1) |
| NE | 59 | 159 | 105 | 323 | 0.18 | 0.49 | **0.33** | 0.7 | 0.99 | **1.34** |
| NW | 223 | 613 | 284 | 1120 | 0.20 | 0.55 | 0.25 | 0.76 | 1.10 | 1.04 |
| SE | 53 | 109 | 60 | 222 | 0.24 | 0.49 | 0.27 | 0.91 | 0.99 | 1.11 |
| SW | 473 | 720 | 328 | 1521 | **0.31** | 0.47 | 0.22 | **1.19** | 0.95 | 0.89 |
| XX | 242 | 391 | 198 | 831 | 0.29 | 0.47 | 0.24 | 1.11 | 0.95 | 0.98 |
| total | 1050 | 1992 | 975 | 4017 | 0.26 | 0.50 | 0.24 | | | |

The table gives the counts of days with low numbers of ATAAD events (-1), large numbers of ATAAD events (+1), and medium numbers of ATAAD events (0) by large-scale wind direction (NO: northeast, NW: northwest, SO: southeast, SW: southwest, XX: heterogeneous) based on objective weather type classification data. Additionally, respective proportions, $P_c$, are given and the ratio factor of the respective class by the total proportion, $P_c/P$ (where $P$ is the total proportion for all days).

is associated with winds from the northeast (NE): As for days with winds from NE, the proportion is 0.33, whereas, in total, the proportion is 0.24. So, during weather patterns when NE winds prevail, the proportion is by a factor 1.34 greater than for any wind direction. Interpreting this ratio factor as a probability, it suggests that the probability of a large number of ATAAD events occurring was increased by 34%. On the other hand, for days with a wind direction from southwest (SW), the probability of occurrence of a low number of ATAAD events was increased by 19%. Using a Chi-squared test for the difference of two proportions, these differences are significant (NE: $\chi^2 = 10.85$, $df = 1$, $p < 0.001$. SW: $\chi^2 = 13.61$, $df = 1$, $p < 0.001$). There is no such statement possible for the other wind directions.

Also, there is a significant association with the humidity factor as defined in the objective weather classification. For days with the humidity factor showing a dry troposphere, the proportion of days with a low number of ATAAD events was 0.28, so the number of days is increased by the factor 1.15 (Table 3), so ATAAD events occurred 15% more frequently. For days with the humidity factor showing a wet troposphere, the proportion of days with many ATAAD events was 0.31, and the ratio factor was 1.18, so ATAAD events occurred 18% more frequently. Both differences are significant (one-sided Chi-squared test for a dry troposphere: $\chi^2 = 9.89$, $df = 1$, $p < 0.001$; for a wet troposphere: $\chi^2 = 13.28$, $df = 1$, $p < 0.001$). 63.7% of all days with a large number of ATAAD events were dry days, and 52.5% of those days with a low number of ATAAD events were wet days. For Central Europe, flow patterns from southwest are generally associated with changing weather, warm and increasing temperatures, and humid conditions, while flow patterns from northeast are connected with less lively weather and mostly dry air masses. For the regarded period, southwest patterns prevailed for 37.9% and northeast patterns for 8.0%. 10.8% of all days with a large number of ATAAD cases are connected to northeast patterns, and 45.0% of all days with a lower number of ATAAD days

**Table 3. Contingency table comparing frequency of ATAAD events and humidity content of the troposphere.**

| | Number of days | | | total | Proportion $P_c$ | | | Ratio factor $P_c/P$ | | |
|---|---|---|---|---|---|---|---|---|---|---|
| | low (-1) | medium (0) | large (+1) | total | low (-1) | medium (0) | large (+1) | low (-1) | medium (0) | large (+1) |
| FF | 551 | 886 | 354 | 1791 | **0.31** | 0.49 | 0.20 | **1.18** | 1.00 | 0.81 |
| TT | 499 | 1106 | 621 | 2226 | 0.22 | 0.50 | **0.28** | 0.86 | 1.00 | **1.15** |
| total | 1050 | 1992 | 975 | 4017 | 0.26 | 0.50 | 0.24 | | | |

The table gives the same information as in Table 2 but for humidity (FF: wet, TT: dry).

**Table 4. Bio-synop classes (see subsection Bio-synop classes in Materials and methods) at which significantly more or less ATAAD events occur.**

| days | relevant bio-synop classes: (values: *class_intensity_thermal stress*) | in words |
|---|---|---|
| with significantly more AD events | • 4_1_ + k2 / k1 / 00 / w1<br>• 4_2_ + k1 / 00 / w1<br>• 3_1_ + k2 / k1 / 00<br>• 3_2_ + k1 / 00 | More ATAAD events occur after cold fronts (4_) and in the center of cyclones (3_) predominantly with cold stress (k) or no thermal stress (00). |
| with significantly fewer ATAAD events | • 2_1_ + 00 / w1 / w2 / w3 / w4<br>• 2_2_ + 00 / w1 / w2 / w3 / w4<br>• 3_1_ + w1 / w2 / w3 / w4<br>• 3_2_ + w2 | Fewer ATAAD events occur ahead of warm fronts (2_) with no thermal stress (00) or heat stress (w) as well as in cyclone center (3_) with heat stress (w). |

were connected to southwest patterns. There is no significant association for the other wind patterns (northwest, southeast, and those without a prevailing wind direction).

## Association of ATAAD events and bio-synop classes

Between 2006 and 2019, 5113 days with predicted bio-synop classes and observed ATAAD events were analyzed for the region of Berlin and Brandenburg. It was found that significantly more ATAAD events (Chi-squared test for the difference of two proportions, $\chi^2 = 3.27$, $p = 0.035$) occurred during bio-synop classes with cold air advection and in the center of a cyclone including conditions with cold stress or thermal comfort (see section "Data"). Compared to the overall proportion of days with ATAAD events (0.172), the proportion of days with ATAAD events for these bio-synop classes (0.195) was increased by 13%, which means that ATAAD events occurred 13% more frequently during these identified bio-synop classes. In contrast, significantly fewer ATAAD events (Chi-squared test, $\chi^2 = 6.55$, $p = 0.005$) occurred during bio-synop classes with warm air advection and in the center of a cyclone with thermal stress (see Table 4). Compared to the overall proportion of days with ATAAD events (0.172), the proportion of days with ATAAD events for these bio-synoptic classes (0.139) was 19% smaller, which means that ATAAD events occurred 19% less frequently during these identified bio-synop classes. These findings reveal that during certain meteorological conditions (corresponding to the bio-synop classes 4 and 3 with cold stress; for more details, see Table 4), the occurrence of ATAAD events was increased. These bio-synop classes have the advantage that they take into account warm and cold fronts, and they are known to have an impact on the health of patients with hypertension. Therefore, the results reconfirm and refine the findings of previous studies that hypertension is a significant risk factor for aortic dissection by revealing that during weather conditions that have a negative impact on the health of patients with hypertension, the occurrence of ATAAD events is significantly increased. The results also exhibit weather conditions with significantly fewer ATAAD events (Table 4).

## Discussion

The analysis results in this study underpin the hypothesis that there is a connection between certain meteorological conditions and the occurrence of ATAAD events in Germany. There is a positive association between ATAAD events and those weather patterns that favor hypertension. Cold temperatures are a risk factor for high blood pressure due to vasoconstriction, and high blood pressure is a risk factor for ATAAD events (see Introduction section). This might explain the seasonal patterns observed with a distinct seasonal pattern of ATAAD events corresponding to minima in summer and maxima in autumn/winter/spring (section "Seasonal

patterns"). The pattern resembles the absolute value function of cosine. Such seasonal patterns, including a higher number of ATAAD events in winter and fewer ATAAD events in summer, have previously been determined in several earlier studies [9–16, 20–22, 24] as well as in review studies [17, 19, 23]. [18] found the highest occurrence of ATAAD events in spring and merely the second highest in winter and as well the lowest in summer. Some studies and also one review study [32] already focussed on certain meteorological parameters, especially on air temperature as well as less frequently on air pressure, among others, and found an association with ATAAD events [14, 15, 20, 24, 27–31, 33–35].

The sub-study using the bio-synop classes for the German region of Berlin and Brandenburg (see section "Association of ATAAD events and bio-synop classes") with similar characteristics of meteorological parameters but grouped to classes reveals even more details and new findings, which have not yet been described previously. Significantly, more ATAAD events occurred during bio-synop classes with cold air advection (on the back of cold fronts) as well as in the center of a cyclone during conditions with cold stress or thermal comfort. On the other hand, significantly fewer ATAAD events occurred during bio-synop classes with warm air advection (ahead of warm fronts) and in the center of a cyclone with thermal stress. These findings are in conjunction with the observed seasonal ATAAD patterns and the findings from other studies focusing on certain meteorological parameters. For example, the bio-synop class 4 describing the back of cold fronts includes decreasing temperature and simultaneously increasing air pressure. The bio-synop classes group the meteorological parameters to classes with similar changes, and thus, they reveal short-term impacts of changes in weather conditions related to weather fronts and not only long-term or seasonal effects. Related short-term impacts have also been observed based on the single parameter mean temperature during sharp temperature drops between neighboring days in China [35].

To minimize uncertainties that may arise because the bio-synop classes are operational predictions and cold fronts, for example, may move faster or slower than predicted, we used the latest available forecasts (the predicted bio-synop classes for the second half of the current day). Thus, prediction errors are as minor as possible. [35] also found a higher risk of ATAAD occurrences on the concurrent day and attenuated on lag 1 day. After that, their results became insignificant. This corresponds to our investigations with the bio-synop classes of the current day used together with the ATAAD events of the following 24 hours. All in all, the results demonstrate that the bio-synop classes are a valuable tool for analyzing patterns of ATAAD events shorter than seasonal scale in a range of a few days, and, with further research, they also offer the possibility of predicting an increased or decreased risk for the occurrence of ATAAD cases.

The analysis of the data for total Germany confirmed in accordance with other studies that on days with a high number of ATAAD events, on average, the temperatures were lower (by about 4.8 K), but also the water vapor pressure (by about 2.6 hPa). This corresponds with large-scale weather patterns and has been examined by analyzing the predominant weather patterns over Germany. A statistical relationship was found for weather patterns with prevailing winds from the northeast (see section "Association between ATAAD events and weather patterns") with a higher proportion (by about 34%) of days with a large number of ATAAD events occurring. Though it should be critically noted that the weather patterns describe large-scale processes and small-scale processes such as weather fronts may not be adequately reflected in the patterns of the objective weather classification but might be important factors in triggering ATAAD events, as seen in the results of the analyzed bio-synop classes. Generally, a northeastern circulation regime is connected to the transport of continental cooler and drier air masses from East Europe and Russia. On the contrary, the temperatures and the humidity were higher on days with a low number of ATAAD events. This corresponds to large-scale

weather patterns with the advection of warm and humid air masses from the southwest originating from the Mediterranean region.

These results exhibit the hypothesis that weather patterns favoring hypertension also favor ATAAD events.

The growing understanding of these associations between the occurrence of ATAAD events and prevailing weather conditions might allow for creating prediction models, e.g., based on machine learning [70], and for releasing warnings of people under risk when weather conditions possibly favoring ATAAD events are likely to occur. Class labels must be known for such a method, including individual health characteristics and weather conditions. It is crucial to further investigate weather and health conditions favoring ATAAD events to transfer mature prediction tools in clinical practice.Moreover, climate change and future temperature-related cardiovascular impacts, as well as urbanization, air quality, and population aging, have to be considered during further investigations [71–73].

Future investigations might enable a more adjuvant stratification of the risk of suffering from ATAAD to prevent misdiagnosis. The forecasts for humans suffering from pollen allergies or for weather-sensitive individuals (cardiovascular diseases, asthma, rheumatism, wellbeing) distributed by the DWD could serve as a template. The prediction tools to predict the probability of the occurrence of ATAAD events should include not only the health conditions of individuals but also defined weather parameters. This could contribute to reducing misdiagnosed cases of ATAAD and saves time for the further course, which is determining for the patient and his health and life.

## Conclusion

The findings are in conformity with previous studies and reveal an association between the occurrence of ATAAD events and meteorological impacts. For Germany, it was found that significantly more ATAAD events occur during lower temperatures, lower humidity, and prevailing wind patterns from the northeast as well as in autumn/ winter/ spring and during weather conditions with cold air advection on the back of cold fronts throughout the year as well as conditions in the center of a cyclone with prevailing cold stress or thermal comfort. On the contrary, weather conditions with significantly decreased occurrence of ATAAD events have also been identified. Patients with hypertension constitute a risk group for ATAAD events. Based on these research findings, forecasts for a higher risk for ATAAD events depending on the weather conditions might be developed in the future.

## Acknowledgments

We want to thank Melanie Scheller from the Research Data Center of the Federal Statistical Office for her appreciated help regarding the data sets. In addition, we want to gratefully thank Rainer Behrendt from Wetter3.de for his support with the reanalysis data and Bernhard Mühr from Karlsruhe Institute of Technology for the valuable discussions and suggestions.

[59] was downloaded from the Copernicus Climate Change Service (C3S) [58].

The results contain modified Copernicus Climate Change Service information 2023. Neither the European Commission nor ECMWF is responsible for any use that may be made of the Copernicus information or data it contains.

## Author Contributions

**Conceptualization:** Stephan Dominik Kurz, Holger Mahlke, Kathrin Graw, Volkmar Falk, Christoph Knosalla, Andreas Matzarakis.

**Data curation:** Holger Mahlke, Kathrin Graw, Paul Prasse.

**Formal analysis:** Holger Mahlke, Paul Prasse.

**Funding acquisition:** Stephan Dominik Kurz, Volkmar Falk.

**Methodology:** Stephan Dominik Kurz, Holger Mahlke, Kathrin Graw, Christoph Knosalla.

**Project administration:** Stephan Dominik Kurz, Holger Mahlke, Kathrin Graw, Christoph Knosalla, Andreas Matzarakis.

**Supervision:** Stephan Dominik Kurz.

**Validation:** Stephan Dominik Kurz, Holger Mahlke, Kathrin Graw.

**Visualization:** Stephan Dominik Kurz, Holger Mahlke, Kathrin Graw, Andreas Matzarakis.

**Writing – original draft:** Stephan Dominik Kurz, Holger Mahlke, Kathrin Graw.

**Writing – review & editing:** Stephan Dominik Kurz, Holger Mahlke, Kathrin Graw, Paul Prasse, Volkmar Falk, Christoph Knosalla, Andreas Matzarakis.

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
