## [Decision Letter · Decision Letter 0]

10 Oct 2023

PONE-D-23-27232Patterns in acute aortic Dissection and a Connection to meteorological Conditions in GermanyPLOS ONE

Dear Dr. Kurz,

Thank you for submitting your manuscript to PLOS ONE. After careful consideration, we feel that it has merit but does not fully meet PLOS ONE’s publication criteria as it currently stands. Therefore, we invite you to submit a revised version of the manuscript that addresses the points raised during the review process.

We look forward to receiving your revised manuscript.

Kind regards,

Venkatramanan Senapathi, Ph.D.

Academic Editor

PLOS ONE

Journal Requirements:

2. Thank you for stating the following financial disclosure: "NO"

3. Thank you for stating the following in your Competing Interests section: "NO"

Reviewers' comments:

Reviewer's Responses to Questions

**Comments to the Author**

1. Is the manuscript technically sound, and do the data support the conclusions?

Reviewer #1: Yes

Reviewer #2: Yes

2. Has the statistical analysis been performed appropriately and rigorously? 

Reviewer #1: Yes

Reviewer #2: Yes

3. Have the authors made all data underlying the findings in their manuscript fully available?

Reviewer #1: Yes

Reviewer #2: Yes

4. Is the manuscript presented in an intelligible fashion and written in standard English?

Reviewer #1: Yes

Reviewer #2: Yes

5. Review Comments to the Author

Reviewer #1: 1. The authors need to explain in more detail the mechanism of influence between blood pressure and the diseases studied

2. The author's conclusion seems too simple and should continue to distill the key information

3.The author's summary of published results seems to be incomplete. There is a paper designed to design the relationship between cardiovascular and cerebrovascular diseases and weather for the author's reference:

Projections of future temperature-related cardiovascular mortality under climate change, urbanization and population aging in Beijing, China. Environment International 163 (2022) 107231

Estimating the mortality burden attributable to temperature and PM2.5 from the perspective of atmospheric flow. Environment Research Letters,2020, 15,124059.

Impacts of urbanization on the temperature-mortality relationship in Beijing, China. Environmental Research 191 (2020) 110234.

Reviewer #2: Summary and major comments

This study analyzes the seasonal variation of the acute type A aortic dissection (ATAAD) events in Germany and the possible connections between ATAAD and various weather conditions. The authors found an absolute value of cosine-like seasonal variation pattern with significantly fewer ATAAD events in the summer months. They then demonstrated that significantly more ATAAD events occurred during lower temperatures, lower specific humidity, and prevailing wind patterns from the northeast. Furthermore, they identified a positive association between ATAAD events and those weather patterns that favor hypertension. Overall, their results reconfirm and refine the findings of some previous studies.

This study represents an original contribution to our understanding of weather impacts on human health. The manuscript is well-organized and well-written. The conclusions are generally consistent with the stated objective. There is still some room for further improvement. My main concerns are 1) the reason for choosing the CFS reanalysis for this study was not given; 2) the relationship between specific humidity and temperature was not well explained; and 3) the implication of climate change toward a warmer future was not addressed; see the following specific comments for further details. Therefore, my recommendation is to go through some minor revisions before accepting for publication.

Specific comments and suggestions

Page 1, the 3rd line of Abstract: Consider changing “cosine-like pattern” to “cosine-like seasonal variation pattern”.

Page 1, last line: Is “und” a typo for “and”?

L34: By “positive association”, do you mean “significant correlation” or “positive correlation”? It seems to me that temperature and blood pressure is negatively correlated.

L52: You may need to define “CFS” here.

L85: You may need to explain why the CFS reanalysis data are chosen and where they were obtained. The reanalysis only covers the period from 1979 to 2011 at this source: https://www.ncei.noaa.gov/access/metadata/landing-page/bin/iso?id=gov.noaa.ncdc:C00765. From another source at: https://climatedataguide.ucar.edu/climate-data/climate-forecast-system-reanalysis-cfsr, the years of record is from 1979-01 to 2017-11. So I am not sure if CFSR is good enough for this study. Another key limitation of CFSR is that relatively few evaluations of CFSR have been conducted so the performance is not well-known. In my opinion, the ECMWF Reanalysis v5 (ERA5, https://www.ecmwf.int/en/forecasts/dataset/ecmwf-reanalysis-v5) would be better than CFSR for this study. It covers the period from January 1940 to present, at an hourly basis.

L91: I am not sure how this line is connected to the previous line. It is obvious not connected to line 90 in the previous page. There is something wrong here.

L118, L135: Could you provide at least one reference for the Wald-Wolfowitz runs test? A reference for the Wilcoxon rank sum test would be good, too.

L157: Add “on average” after “this is”.

L159-160: Change “approx.” to approximately”, and consider changing “does not” to “may not”. We are not 100% sure if there is a weak increase in the number of ATAAD cases. I suggest that you also plot the corresponding time series of area-averaged temperature in Germany as a subplot in Figure 1. It is understood that the global warming implies an increase in long-term global-averaged temperatures. But it could be possible that a regional mean temperature trend is negative for this short period, e.g., due to some decadal oscillation factor. Either a warming or cooling trend would be worthy of plotting and explaining it.

L207-219: The mid-latitude temperatures in Northern Hemisphere are highly correlated with wind directions. NE winds are often related to low temperatures, and SW winds often bring in warm weather. This relationship is mentioned in the following paragraph. It would be worth briefly mentioning it either at the beginning or the end of this paragraph.

L220-238: It is hard to understand the humidity factor independently. If you use specific humidity (SH) to measure the humidity factor, you can always expect lower SH with lower temperature, and higher SH with higher temperature. Given this positive relation, you may also need to analyze the humidity factor using relative humidity, and compare the result to that of specific humidity. If you want to discuss the humidity factor in the troposphere, there could be a short-lived greenhouse effect of the trapped moisture in the troposphere that could form a positive feedback mechanism leading to a warmer condition (Mo et al., 2022, Commun. Earth Environ., 3, 127, https://doi.org/10.1038/s43247-022-00459-w).

L306: In meteorology, “sub-seasonal” usually means 15 to 75-day variation. Short-range (medium-range) forecasting usually covers the following 1 to 3 days (4 to 14 days).

L318: By “lower-scale”, do you mean “smaller-scale”?

L342: It would be appropriate to add a paragraph here to briefly discuss the implication of climate change to the ATAAD events in the future, given the strong negative correlation between ATAAD and temperatures.

Ruping Mo (ruping.mo@ec.gc.ca)

Senior Research Meteorologist

National Laboratory-West, Environment and Climate Change Canada

Vancouver, BC, Canada

6. PLOS authors have the option to publish the peer review history of their article (what does this mean?). If published, this will include your full peer review and any attached files.

Reviewer #1: No

Reviewer #2: **Yes: **Ruping Mo

---

## [Author Response · Author response to Decision Letter 0]

6 Dec 2023

Reviewer #1: 

1. The authors need to explain in more detail the mechanism of influence between blood pressure and the diseases studied

Our reply: Thanks for your comment. We added the mechanism in more detail at line 38.

2. The author’s conclusion seems too simple and should continue to distill the key information

Our reply: Thanks for this comment we replaced the sentence starting at line 345. 

3. The author’s summary of published results seems to be incomplete. There is a paper designed to design the relationship between cardiovascular and cerebrovascular diseases and weather for the author’s reference:

Projections of future temperature-related cardiovascular mortality under climate change, urbanization and population aging in Beijing, China. Environment International 163 (2022) 107231

Estimating the mortality burden attributable to temperature and PM2.5 from the perspective of atmospheric flow. Environment Research Letters,2020, 15,124059.

Impacts of urbanization on the temperature-mortality relationship in Beijing, China. Environmental Research 191 (2020) 110234.

Our reply: Thanks for these additional two publications. Now, we included them in the outlook of the discussion after line 334.

Reviewer #2: 

Summary and major comments

This study analyzes the seasonal variation of the acute type A aortic dissection (ATAAD) events in Germany and the possible connections between ATAAD and various weather conditions. The authors found an absolute value of cosine-like seasonal variation pattern with significantly fewer ATAAD events in the summer months. They then demonstrated that significantly more ATAAD events occurred during lower temperatures, lower specific humidity, and prevailing wind patterns from the northeast. Furthermore, they identified a positive association between ATAAD events and those weather patterns that favor hypertension. Overall, their results reconfirm and refine the findings of some previous studies.

This study represents an original contribution to our understanding of weather impacts on human health. The manuscript is well-organized and well-written. The conclusions are generally consistent with the stated objective. There is still some room for further improvement. My main concerns are

1) the reason for choosing the CFS reanalysis for this study was not given;

2) the relationship between specific humidity and temperature was not well explained; and

3) the implication of climate change toward a warmer future was not addressed; see the following specific comments for further details. Therefore, my recommendation is to go through some minor revisions before accepting for publication.

Specific comments and suggestions

Page 1, the 3rd line of Abstract: Consider changing “cosine-like pattern” to “cosine-like seasonal variation pattern”.

Our reply: Thanks, that’s a good suggestion. We have changed it.

Page 1, last line: Is “und” a typo for “and”?

Our reply: Yes, that’s a typo. We have changed it.

L34: By “positive association”, do you mean “significant correlation” or “positive correlation”? It seems to me that temperature and blood pressure is negatively correlated. 

Our reply: Yes, that was incorrect, thanks. We have changed it to “relationship” as a general term, as the details are explained in the following sentences. 

L52: You may need to define “CFS” here.

Our reply: We have changed this to ERA5, see next comment (below). 

L85: You may need to explain why the CFS reanalysis data are chosen and where they were obtained. The reanalysis only covers the period from 1979 to 2011 at this source: https://www.ncei.noaa.gov/access/metadata/landing-page/bin/iso?id=gov.noaa.ncdc:C00765. From another source at: https://climatedataguide.ucar.edu/climate-data/climate-forecast-system-reanalysis-cfsr, the years of record is from 1979-01 to 2017-11. So I am not sure if CFSR is good enough for this study. Another key limitation of CFSR is that relatively few evaluations of CFSR have been conducted so the performance is not well-known. In my opinion, the ECMWF Reanalysis v5 (ERA5, https://www.ecmwf.int/en/forecasts/dataset/ecmwf-reanalysis-v5) would be better than CFSR for this study. It covers the period from January 1940 to present, at an hourly basis.

Our reply: Thank you for recommending the Era5 data. After consideration, we agreed to your concerns about the CFS and accepted your suggestion to use ERA5 reanalyses data instead. We recalculated the results again with Era5 and decided to use these data for this publication instead of the CFSR data. Nevertheless, the results are similar (more ATAAD events during lower temperatures and dryer conditions). So, we have made the required changes in the database section, the tables, and the Acknowledgements section. 

L91: I am not sure how this line is connected to the previous line. It is obvious not connected to line 90 in the previous page. There is something wrong here.

Our reply: Thanks for this comment. We revised the sentence. It’s hopefully clearer, now.

L118, L135: Could you provide at least one reference for the Wald-Wolfowitz runs test? A reference for the Wilcoxon rank sum test would be good, too.

Our reply: Okay, sure. We have added references to both statistical tests.

L157: Add “on average” after “this is”.

Our reply: We have added this, thanks.

L159-160: Change “approx.” to approximately”, and consider changing “does not” to “may not”. We are not 100% sure if there is a weak increase in the number of ATAAD cases. I suggest that you also plot the corresponding time series of area-averaged temperature in Germany as a subplot in Figure 1. It is understood that the global warming implies an increase in long-term global-averaged temperatures. But it could be possible that a regional mean temperature trend is negative for this short period, e.g., due to some decadal oscillation factor. Either a warming or cooling trend would be worthy of plotting and explaining it. 

Our reply: Thank you for this suggestion. We have changed “approx.” and made the change to “may not”.

We plotted the mean temperature trend for this short period (see below plot) but we couldn’t find a significant trend. Therefore we decided not to add a subplot but have added a sentence at the end of line 157.

L207-219: The mid-latitude temperatures in Northern Hemisphere are highly correlated with wind directions. NE winds are often related to low temperatures, and SW winds often bring in warm weather. This relationship is mentioned in the following paragraph. It would be worth briefly mentioning it either at the beginning or the end of this paragraph.

Our reply: Yes, good idea, thanks. We added it at the beginning of the paragraph after the first sentence at line 208.

L220-238: It is hard to understand the humidity factor independently. If you use specific humidity (SH) to measure the humidity factor, you can always expect lower SH with lower temperature, and higher SH with higher temperature. Given this positive relation, you may also need to analyze the humidity factor using relative humidity, and compare the result to that of specific humidity. If you want to discuss the humidity factor in the troposphere, there could be a short-lived greenhouse effect of the trapped moisture in the troposphere that could form a positive feedback mechanism leading to a warmer condition (Mo et al., 2022, Commun. Earth Environ., 3, 127, https://doi.org/10.1038/s43247-022-00459-w). Spec. humidity raus!

Our reply: We are not sure if we caught your comment, but, now on base of the Era5 data, we calculated different humidity parameters: specific humidity, relative humidity, vapor pressure, and dewpoint temperature.

However, for our conduction, we need an absolute humidity measure. Therefore we initially (first submission) chose the specific humidity. It is also possible to choose the vapor pressure which is a common humidity measure in Human-Biometeorology, and we decided to discuss the vapor pressure for the revision.

The result that more ATAAD events occur during dryer conditions could be achieved with the specific humidity, the vapor pressure, and the dewpoint temperature. Using the relative humidity, we got the result that days with many ATAAD events have a higher relative humidity, especially in Winter. But this is not what we wanted to point out. The relative humidity only assesses the humidity in the air related to the temperature. With the vapor pressure, we can make a statement about the absolute moisture content and that during dryer conditions more ATAAD events occur. 

However, humidity is one of many different parameters affecting human health, especially patients with hypertonia. The mechanism is not yet fully understood. Some studies found, that blood vessels expand due to high humidity. The expansion of blood vessels causes the blood pressure to decrease and therefore high humidity may have a positive effect on hypertension. But the vegetative nervous system doesn’t react only on one parameter but on many environmental stresses simultaneously. Therefore, human health is always dependent on multifactorial impacts and more studies are required to determine the impact of humidity on human health and especially on patients with hypertonia.

L306: In meteorology, “sub-seasonal” usually means 15 to 75-day variation. Short-range (medium-range) forecasting usually covers the following 1 to 3 days (4 to 14 days).

Our reply: Yes, thanks for this explanation. We deleted "sub-seasonal" and changed it to “for analyzing patterns of ATAAD events shorter than seasonal scale in a range of a few days”.

L318: By “lower-scale”, do you mean “smaller-scale”?

Our reply: Yes, thanks, we changed it accordingly.

L342: It would be appropriate to add a paragraph here to briefly discuss the implication of climate change to the ATAAD events in the future, given the strong negative correlation between ATAAD and temperatures.

Our reply: Yes, thanks, we now addressed climate change after line 334.

---

## [Decision Letter · Decision Letter 1]

19 Dec 2023

Patterns in acute aortic Dissection and a Connection to meteorological Conditions in Germany

PONE-D-23-27232R1

Dear Dr. Kurz,

We’re pleased to inform you that your manuscript has been judged scientifically suitable for publication and will be formally accepted for publication once it meets all outstanding technical requirements.

Kind regards,

Venkatramanan Senapathi, Ph.D.

Academic Editor

PLOS ONE

Additional Editor Comments (optional):

Reviewers' comments:

Reviewer's Responses to Questions

**Comments to the Author**

1. If the authors have adequately addressed your comments raised in a previous round of review and you feel that this manuscript is now acceptable for publication, you may indicate that here to bypass the “Comments to the Author” section, enter your conflict of interest statement in the “Confidential to Editor” section, and submit your "Accept" recommendation.

Reviewer #1: (No Response)

Reviewer #2: All comments have been addressed

2. Is the manuscript technically sound, and do the data support the conclusions?

Reviewer #1: (No Response)

Reviewer #2: Yes

3. Has the statistical analysis been performed appropriately and rigorously? 

Reviewer #1: (No Response)

Reviewer #2: Yes

4. Have the authors made all data underlying the findings in their manuscript fully available?

Reviewer #1: (No Response)

Reviewer #2: Yes

5. Is the manuscript presented in an intelligible fashion and written in standard English?

Reviewer #1: (No Response)

Reviewer #2: Yes

6. Review Comments to the Author

Reviewer #1: (No Response)

Reviewer #2: The authors have addressed all my concerns with the original manuscript. This revised manuscript is ready for publication.

7. PLOS authors have the option to publish the peer review history of their article (what does this mean?). If published, this will include your full peer review and any attached files.

Reviewer #1: No

Reviewer #2: **Yes: **Ruping Mo

---

## [Editor Report · Acceptance letter]

2 Jan 2024

PONE-D-23-27232R1 

PLOS ONE

Dear Dr. Kurz, 

I'm pleased to inform you that your manuscript has been deemed suitable for publication in PLOS ONE. Congratulations! Your manuscript is now being handed over to our production team.

Kind regards, 

on behalf of

Dr. Venkatramanan Senapathi 

Academic Editor

PLOS ONE